# Bridging the Gap from Asymmetry Tricks to Decorrelation Principles in Non-contrastive Self-supervised Learning

**Kang-Jun Liu[1]    Masanori Suganuma[1]    Takayuki Okatani[1,2]**
[1]Graduate School of Information Sciences, Tohoku University
[2]RIKEN Center for AIP
`{kjliu, suganuma, okatani}@vision.is.tohoku.ac.jp`

## Abstract

Recent non-contrastive methods for self-supervised representation learning show promising performance. While they are attractive since they do not need negative samples, it necessitates some mechanism to avoid collapsing into a trivial solution. Currently, there are two approaches to collapse prevention. One uses an asymmetric architecture on a joint embedding of input, e.g., BYOL and SimSiam, and the other imposes decorrelation criteria on the same joint embedding, e.g., Barlow-Twins and VICReg. The latter methods have theoretical support from information theory as to why they can learn good representation. However, it is not fully understood why the former performs equally well. In this paper, focusing on BYOL/SimSiam, which uses the stop-gradient and a predictor as asymmetric tricks, we present a novel interpretation of these tricks; they implicitly impose a constraint that encourages feature decorrelation similar to Barlow-Twins/VICReg. We then present a novel non-contrastive method, which replaces the stop-gradient in BYOL/SimSiam with the derived constraint; the method empirically shows comparable performance to the above SOTA methods in the standard benchmark test using ImageNet. This result builds a bridge from BYOL/SimSiam to the decorrelation-based methods, contributing to demystifying their secrets. Source code is available at `https://github.com/KJ-rc/bridging-the-gap`.

## 1   Introduction

Recently, many methods have been proposed for self-supervised learning of visual representation [1–6, 8–10, 12, 16] They share a fundamental idea: to learn a visual representation of images invariant to a range of image transformations maintaining their semantics. This idea is implemented as the optimization of an objective that the joint embeddings of different *views* of an input image, e.g., different image subregions cut from a single image and subjected to additional data augmentation, should be close to each other.

The methods are categorized into contrastive and non-contrastive methods. The latter needs only positive samples (i.e., different views of the same images), whereas the former also needs negative samples (i.e., different images). Therefore, although it stabilizes learning, contrastive methods tend to need large memory [4] or additional measures (e.g., a momentum encoder [12]) to alleviate it. On the other hand, the non-contrastive methods are potentially more efficient. However, their objective has a trivial solution: to map all inputs into a single point in the feature space. Thus, non-contrastive methods need additional measures to prevent this feature collapse.

To do so, BYOL/SimSiam [6, 10] employs asymmetric structure into the joint embeddings, i.e., stop-gradient and a predictor; see Fig. 2(a). The predictor is a subnetwork placed on top of one of the two

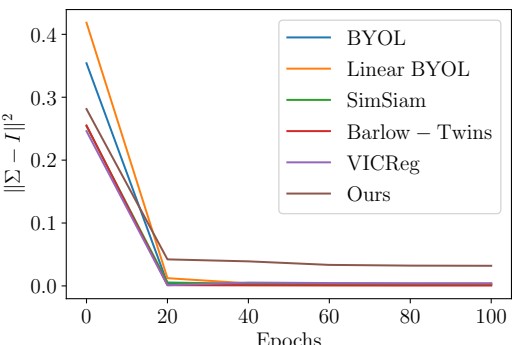

Figure 1: Experimental results verify our claim that asymmetry tricks implicitly encourage feature decorrelation. $\Sigma$ is the correlation matrix of features extracted from ImageNet validation images. BYOL/SimSiam makes $\Sigma$ approach to an identity matrix $I$, so do feature decorrelation methods (i.e., Barlow-Twins/VICReg). This is also the case with the proposed method. See Sec. 5.2 for more details.

joint-embedding pipelines; it predicts the output of the other pipeline from its input. BYOL/SimSiam uses the loss of minimizing the prediction error, aiming at the above objective of learning invariant feature representation. It back-propagates the gradient only to the first pipeline having the predictor while it stops the gradient flow to the second pipeline. (While BYOL inherited a momentum encoder for the second pipeline, later it was recognized [6, 14] that it is not indispensable.)

However, it has not been well understood why the above asymmetric structure prevents collapse and further enables to learn good representation. Tian et al. theoretically tackle this question, deriving some results helping to understand the tricks [14]. Nonetheless, we still lack a complete understanding of the working mechanism of the asymmetry tricks, especially why they lead to the learning of good representation.

In this paper, we show that the asymmetry tricks employed by BYOL/SimSiam have an additional implicit effect, such that the features extracted from different images will be decorrelated. Specifically, we extend Tian et al.'s analyses under similar assumptions. We then show that the updating dynamics of the predictor's weights and the features (i.e., the predictor's inputs) indicates that the minimization implicitly imposes a constraint achieving feature decorrelation; see Fig. 1. Finally, we show through experiments that we can eliminate stop-gradient by using the constraint as an explicit regularizer with the standardization of features. Specifically, using the regularizer and the standard invariance loss leads to learning as good representation as the state-of-the-art SSL methods, including BYOL/SimSiam.

This result provides a link from BYOL/SimSiam to the other group of non-contrastive methods, i.e., Barlow-Twins [16] and VICReg [1]. These methods do not employ an asymmetric structure and instead incorporate an explicit objective of decorrelating features of different input images. As explained in [16], information theory supports the goodness of feature decorrelation for representation learning, explaining why it helps learn a good representation. The link between the two groups of methods implies that the same is true for BYOL/SimSiam. In short, it implies that the asymmetry tricks implicitly achieve feature decorrelation, leading to the learning of good representation without collapse.

## 2 Related Work

### 2.1 Asymmetry Tricks: BYOL/SimSiam

BYOL [10] and SimSiam [6] employ asymmetric architectures; see Fig. 2(a). Let $f^1$, $f^2 \in \mathbb{R}^d$ be the outputs of an identical network for two views of an identical image $x$. BYOL minimizes the following loss:

$$\mathcal{L}_{\text{BYOL}} = \mathbb{E}_x \left[ \left\| \frac{p(f^1)}{\|p(f^1)\|_2} - \text{StopGrad}\left(\frac{f^2}{\|f^2\|_2}\right) \right\|_2^2 \right], \tag{1}$$

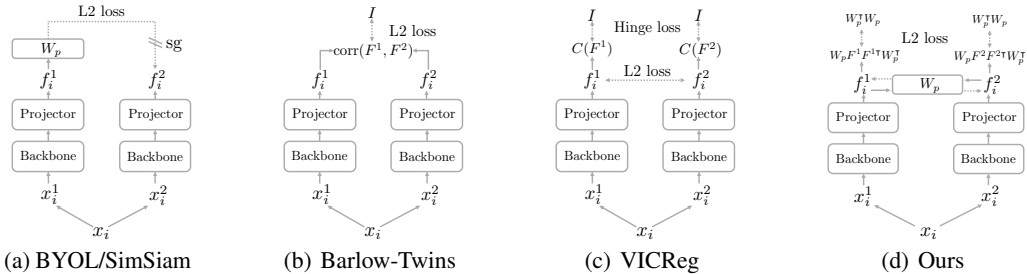

(a) BYOL/SimSiam     (b) Barlow-Twins     (c) VICReg     (d) Ours

Figure 2: Illustrations of representative non-contrastive self-supervised methods for representation learning and our method.

where $p(\cdot)$ is a predictor, e.g., a two-layer MLP with ReLU and batch normalization in the intermediate layer; and $\mathrm{StopGrad}$ indicates that the gradients are not back-propagated with the second path for $f^2$. There are several differences between BYOL and SimSiam; while BYOL uses a momentum encoder to compute $f^2$, SimSiam does not, and SimSiam uses a loss based on cosine similarity.

It is unclear why BYOL and SimSiam can avoid collapsing to a trivial solution even though they only impose augmentation invariance. In [14], incorporating some simplification of the model and assumptions on the data, the authors analyze the dynamics of how the above minimization updates the network weights. They then show that i) there is an implicit balancing effect between the projector's weights and the predictor's weight, which may help prevent collapse; ii) stop-gradient is indispensable for collapse prevention. They then present a method named DirectPred, which directly sets the weight of a linear predictor based on the principal component analysis of the extracted features. However, these do not fully explain the working mechanism of BYOL/SimSiam, especially why they can learn good representation.

## 2.2 Decorrelation-based: Barlow-Twins/VICReg

As shown in Fig. 2(b), Barlow-Twins [16] minimizes the following loss:

$$\mathcal{L}_{\mathrm{BT}} = \|\mathrm{corr}(F^1, F^2) - I\|_{\mathrm{F}}^2, \tag{2}$$

where $F^1(\in \mathbb{R}^{d \times n})$ and $F^2(\in \mathbb{R}^{d \times n})$ are matrices storing all the $f_i^1$'s and $f_i^2$'s of a batch of inputs $x_i$'s $(i = 1, \ldots, n)$, respectively; $\mathrm{corr}(F^1, F^2)(\in \mathbb{R}^{d \times d})$ is the cross-correlation between $F^1$ and $F^2$ along the batch dimension and $I$ is the identity matrix. Note that Zbontar et al. separate the loss into the terms of diagonal and non-diagonal components of $\mathrm{corr}(F^1, F^2)$ with different weighting constants, which is omitted in (2) for brevity.

As shown in Fig. 2(c), VICReg [1] minimizes the following loss:

$$\mathcal{L}_{\mathrm{VICReg}} = \frac{1}{2n} \sum_i^n \|f_i^1 - f_i^2\|_2^2 + \frac{\nu}{d} \sum_{i \neq j}^d (C(F^1)_{ij}^2 + C(F^2)_{ij}^2)$$

$$+ \mu \sum_i^d (\max(0, 1 - \sqrt{C(F^1)_{ii} + \epsilon}) + \max(0, 1 - \sqrt{C(F^2)_{ii} + \epsilon})), \tag{3}$$

where $C(F^1)$ and $C(F^2)$ are the auto-covariance matrices of $f^1$ and $f^2$, respectively, and $\mu$ and $\nu$ are weighting constants. As shown above, VICReg separates the diagonal and non-diagonal components of the matrices and assigns different weights, as in Barlow-Twins [16], and employs a hinge loss for the diagonal term.

Barlow-Twins and VICReg share a similar objective, i.e., decorrelating features of different inputs, although the targets are slightly different (cross-correlation vs. auto-covariance).

## 3 Roles of Asymmetry Tricks

In this section, we consider what role(s) the asymmetry tricks, i.e., a predictor and stop-gradient, play. We extend the results of the study [14], showing a new interpretation.

### 3.1 Problem Statement

We succeed the assumptions/settings employed in [14] except an assumption on the data distribution. Namely, we approximate the backbone + projector (i.e., the mapping from an input image $x$ to its feature $f$) to be a linear mapping. We then consider (a variant of) BYOL with a linear predictor minimizing the following $\ell_2$ loss:

$$\mathcal{L} = \frac{1}{2}\mathbb{E}_x[\|W_\mathrm{p}f^1 - \mathrm{StopGrad}(f^2)\|_2^2], \tag{4}$$

where $\mathbb{E}_x$ is the expectation over the distribution of an input image $x$; $f^1$ and $f^2 (\in \mathbb{R}^d)$ are the outputs of the projector for two different views of $x$; $W_\mathrm{p}$ is the weight of the linear predictor. Note that the original version of BYOL employs a different loss (1), and a nonlinear predictor, i.e., an MLP having two or more layers with ReLUs. We do not use the assumption on $x$'s distribution employed in [14] and instead assume each $x$ to be normalized, i.e., $x^\top x = 1$.

In the implementation of BYOL/SimSiam and others, $\mathbb{E}_x[\cdot]$ is replaced with the average over a mini-batch. Using $F^1 = [f_1^1, \ldots, f_n^1]/\sqrt{n}$ and $F^2 = [f_1^2, \ldots, f_n^2]/\sqrt{n}$, where $n$ is the mini-batch size, we have $F^1 F^{1\top} \approx \mathbb{E}_x[f^1 f^{1\top}]$ etc. As a result, (4) is rewritten as

$$\mathcal{L} \approx \frac{1}{2n}\sum_{i=1}^n \|W_\mathrm{p}f_i^1 - \mathrm{StopGrad}(f_i^2)\|_2^2 = \frac{1}{2}\|W_\mathrm{p}F^1 - \mathrm{StopGrad}(F^2)\|_\mathrm{F}^2. \tag{5}$$

BYOL/SimSiam use not only the orientation $F^1 \to F^2$ but also the other way $F^2 \to F^1$; letting $\mathcal{L}'$ be the loss for the second way, they minimize $\mathcal{L} + \mathcal{L}'$. For brevity, we show only $\mathcal{L}$ in what follows unless otherwise noted.

Finally, our goal is to understand how $W_\mathrm{p}$ are updated and how $F^i$'s behave as a result when minimizing (4), or equivalently (5).

### 3.2 Dynamics of $W_p$ and $F^1$

Following [14], we regard the network weights varying during loss minimization as time-dependent variables, e.g., $W_\mathrm{p} = W_\mathrm{p}(t)$. We do this also for intermediate layer outputs, e.g., $F^1 = F^1(t)$.

**Theorem 3.1.** *When minimizing the loss (5) with weight decay $\eta$, $W_\mathrm{p}(t)$ and $F^1(t)$ satisfy*

$$W_\mathrm{p}(t)^\top W_\mathrm{p}(t) = F^1(t)F^1(t)^\top + e^{-2\eta t}C, \tag{6}$$

*where $C$ is a constant matrix determined by the initial weights.*

To prove Theorem 3.1, we need the following lemmas.

**Lemma 3.2.** *The derivatives of the loss (5) with respect to $W_\mathrm{p}$ and $F^1$ can be respectively given as follows:*

$$\frac{\partial \mathcal{L}}{\partial W_\mathrm{p}} = W_\mathrm{p}F^1 F^{1\top} - F^2 F^{1\top}, \tag{7}$$

$$\frac{\partial \mathcal{L}}{\partial F^1} = W_\mathrm{p}^\top W_\mathrm{p}F^1 - W_\mathrm{p}^\top F^2. \tag{8}$$

**Lemma 3.3.** *When minimizing the loss (5) with weight decay $\eta$ under the above assumptions, $F^1(t)$ is updated with the following velocity:*

$$\dot{F}^1 \equiv \frac{dF^1(t)}{dt} = -\frac{\partial \mathcal{L}}{\partial F^1} - \eta F^1. \tag{9}$$

We show the proof of these lemmas in the supplementary material. Now we prove Theorem 3.1.

*Proof of Theorem 3.1.* When minimizing the loss (5) with gradient descent and weight decay, the velocity of $W_\mathrm{p}$ and $F^1$ during the minimization are respectively given by

$$\dot{W}_\mathrm{p} \equiv \frac{dW_\mathrm{p}(t)}{dt} = -\frac{\partial \mathcal{L}}{\partial W_\mathrm{p}} - \eta W_\mathrm{p}, \tag{10}$$

$$\dot{F}^1 \equiv \frac{dF^1(t)}{dt} = -\frac{\partial \mathcal{L}}{\partial F^1} - \eta F^1. \tag{11}$$

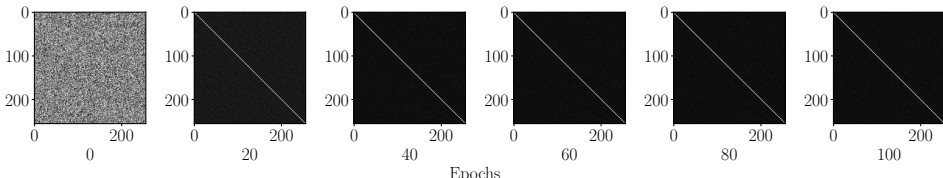

Figure 3: Visualization of $W_{\mathrm{p}} \in \mathbb{R}^{256 \times 256}$ at different epochs of BYOL with a linear predictor. The predictor weight $W_{\mathrm{p}}$ approaches a diagonal matrix $cI$ with a positive value $c$.

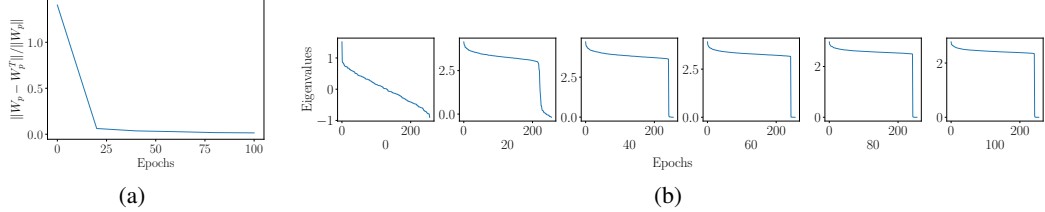

(a)                                                                          (b)

Figure 4: The predictor weight $W_{\mathrm{p}}$ of BYOL with a linear predictor (a) becomes symmetric and (b) its eigenvalues converge to either a positive value or 0, as training epochs increase.

We then calculate $W_{\mathrm{p}}^{\top} \dot{W}_{\mathrm{p}} - \dot{F}^1 F^{1\top}$, to which we substitute (7) and (8), yielding

$$W_{\mathrm{p}}(t)^{\top} \frac{dW_{\mathrm{p}}(t)}{dt} + \eta W_{\mathrm{p}}(t)^{\top} W_{\mathrm{p}}(t) = \frac{dF^1(t)}{dt} F^1(t)^{\top} + \eta F^1(t) F^1(t)^{\top}. \tag{12}$$

Then, adding the above with its transposed version and then taking an integral over $t$, we have

$$e^{2\eta t} W_{\mathrm{p}}(t)^{\top} W_{\mathrm{p}}(t) = e^{2\eta t} F^1(t) F^1(t)^{\top} + C, \tag{13}$$

where $C$ is a constant matrix determined by the initial values $W_{\mathrm{p}}(0)$ and $F^1(0)$. The multiplication with $e^{-2\eta t}$ yields (6). $\qquad\square$

For a large $t$, (6) reduces to
$$W_{\mathrm{p}}(t)^{\top} W_{\mathrm{p}}(t) = F^1(t) F^1(t)^{\top} \tag{14}$$
This means that minimizing (5) makes $W_{\mathrm{p}}^{\top} W_{\mathrm{p}}$ approach to $F^1 F^{1\top} = (1/n) \sum_{i=1}^{n} f_i f_i^{\top}$, the uncentered covariance of $f_1, \ldots, f_n$.

### 3.3 Relation to Feature Decorrelation

Equation (14) indicates the decorrelation of features *if* $W_{\mathrm{p}}(t) W_{\mathrm{p}}(t)^{\top}$ is diagonal. We empirically found this true for BYOL with a linear predictor; $W_{\mathrm{p}}$ quickly converges to a diagonal matrix as $t$ increases, as shown in Fig. 3. This empirical observation together with (14) implies the connection between BYOL/SimSiam and the decorrelation-based methods.

We provide another empirical evidence based on the eigenvalues of $W_{\mathrm{p}}$. Figure 4(a) shows that $W_{\mathrm{p}}$ becomes symmetric with epochs. Figure 4(b) shows that the majority of its eigenvalues approximately approach a single constant $c$ and the others go to zero. (This agrees with a theoretical result in [14] that each eigenvalue of $W_{\mathrm{p}}$ will be either 0 or a positive value at its stable convergence/equilibrium state.) These two observations, i.e., $W_{\mathrm{p}}$ is symmetric and has an identical eigenvalue, indicates that $W_{\mathrm{p}}$ approaches $cI$.

## 4 Deriving a Method without StopGrad

### 4.1 Basic Idea

Aiming to validate the above implication, we consider eliminating $\mathrm{StopGrad}$ from (5) and instead impose (14) explicitly in training. Specifically, we eliminate $\mathrm{StopGrad}$ and incoporate $\|W_{\mathrm{p}}^{\top} W_{\mathrm{p}} -$

$F^1 F^{1\top}\|_{\mathrm{F}}^2$ as a regularization term as

$$\mathcal{L} = \frac{1}{2}\big(\|W_{\mathrm{p}}F^1 - F^2\|_{\mathrm{F}}^2 + \lambda\|W_{\mathrm{p}}^\top W_{\mathrm{p}} - F^1 F^{1\top}\|_{\mathrm{F}}^2\big). \tag{15}$$

The underlying thought is that since the presence of the asymmetric tricks leads to (14) in the analysis of the dynamics, we wish to see if the explicit imposition of (14) can substitute StopGrad, which is the major difference between the two categories of non-contrastive methods. We will show our empirical findings below that this approach works well when used with either the standardization of features (by adding a standardization layer on the projector's output) or the elimination of the predictor. We will refer to the former as STD and the latter as I-PRED (i.e., identity predictor).

## 4.2 Some Consideration for Implementation

BYOL/SimSiam consider the symmetric loss, i.e., the sum of the losses in the both ways (i.e., $F^1 \leftrightarrow F^2$). Incorporating the proposed regularizer in each way, the total loss will be

$$\mathcal{L} = \frac{1}{2}\Big[\big(\|W_{\mathrm{p}}F^1 - F^2\|_{\mathrm{F}}^2 + \|W_{\mathrm{p}}F^2 - F^1\|_{\mathrm{F}}^2\big)$$
$$+ \lambda\big(\|W_{\mathrm{p}}^\top W_{\mathrm{p}} - F^1 F^{1\top}\|_{\mathrm{F}}^2 + \|W_{\mathrm{p}}^\top W_{\mathrm{p}} - F^2 F^{2\top}\|_{\mathrm{F}}^2\big)\Big]. \tag{16}$$

Now, we consider replacing $F^1$ and $F^2$ in the third and fourth terms as $F^1 \to W_{\mathrm{p}}F^2$ and $F^2 \to W_{\mathrm{p}}F^1$, respectively. Assuming the first and second terms are sufficiently small, we can say that this modification will not change the objective here. We found through our experiments that this modification leads to slightly better performance and less sensitivity to the choice of $\lambda$.

Going back to one-way expression (i.e., $F^1 \to F^2$), we write the modified loss by pairing the first and the fourth terms of (16) (and the second and the third terms in the other way) as follows:

$$\mathcal{L} = c_{\mathrm{inv}} \overbrace{\|W_{\mathrm{p}}F^1 - F^2\|_{\mathrm{F}}^2}^{\text{Augmentation Invariance}} + \frac{c_{\mathrm{cov}}}{d} \underbrace{\|W_{\mathrm{p}}^\top W_{\mathrm{p}} - W_{\mathrm{p}}(F^1 F^{1\top})W_{\mathrm{p}}^\top\|_{\mathrm{F}}^2}_{\text{Covariance Maintaining}}, \tag{17}$$

where $c_{\mathrm{inv}}$ and $c_{\mathrm{cov}}$ are weighting constants of the two terms, and $d$ is the feature dimension size (i.e., the column size of $F^1$); the division by $d$ makes the effective range of $c_{\mathrm{cov}}$ similar to $c_{\mathrm{inv}}$.

We summarize the interpretation of the above loss here. The first term enforces augmentation invariance of $f_i^1$ and $f_i^2$ for any $i(=1,\ldots,n)$, and the second term enforces covariance maintaining, or specifically, avoiding the collapse of (uncentered) covariance of $f_1^1, \ldots, f_n^1$.

It should be noted again that the loss (17) does not have a factor directly preventing feature collapse, as with BYOL/SimSiam. An obvious collapsing case is when the projector always outputs zero, i.e., $f_i^1 = f_i^2 = 0$ for any $i$. In this case, the loss vanishes when $W_{\mathrm{p}} = O$. To avoid this, we apply standardization to the output of the projector, i.e., $F^1 = [f_1^1, \ldots, f_n^1]$. Specifically, we compute

$$\bar{f}_i^1 = \frac{f_i^1 - \mu}{\sigma}, \tag{18}$$

where $\mu = \frac{1}{n}\sum_{j=1}^n f_j^1$ and $\sigma^2 = \frac{1}{n-1}\sum_{j=1}^n (f_j^1 - \mu)^2$, and set $F^1 \leftarrow [\bar{f}_1^1, \ldots, \bar{f}_n^1]/\sqrt{n}$. We perform the same for $F^2$. Note that SimSiam [6] employs the same strategy, and interestingly, so does Barlow-Twins; it is reported [1] that Barlow-Twins performs slightly worse without a standardization layer. However, it is also known that standardization alone is not enough; the space of $f^1$ can be degenerate, as reported in [13]. As will be shown later, we empirically found that the loss (17) plus the feature standardization successfully prevents $W_{\mathrm{p}}$ from being rank-deficient.

## 4.3 (Dis)similarity with Other Methods

Our method does not use StopGrad, which is a stark difference from BYOL/SimSiam/Direct-Pred. In this sense, our method may be more similar to decorrelation-based methods, i.e., Barlow-Twins/VICReg.

Table 1: **Results for different combinations** of the number $k$ of partitions, $c_{\text{inv}}$, and $c_{\text{cov}}$ in (19).

| #PART | INV | COV | ACC |
|---|---|---|---|
| 1 | 1 | 1 | 61.4 |
| 1 | 1 | 25 | 60.6 |
| 1 | 25 | 1 | 60.1 |
| 1 | 25 | 25 | 63.4 |
| 8 | 25 | 1 | 67.3 |
| 8 | 25 | 25 | 66.2 |
| 32 | 25 | 1 | 67.3 |

Table 2: **Dependency on the size of the projector.**

| PROJECTOR | #PART | INV | COV | ACC |
|---|---|---|---|---|
| 4096-256 | 1 | 1 | 1 | 54.7 |
| 4096-256 | 32 | 10 | 1 | 63.7 |
| 2048-2048-2048 | 1 | 1 | 1 | 61.4 |
| 2048-2048-2048 | 32 | 10 | 1 | 66.6 |
| 2048-2048-2048 | 32 | 25 | 1 | 67.3 |
| 8192-8192-8192 | 8 | 25 | 8 | 69.0 |

Roughly speaking, we can say that Barlow-Twins enforces the cross-correlation between $F^1$ and $F^2$ to be an identity matrix, and VICReg enforces the auto-covariance of $F^1$ (and of $F^2$) to be an identity matrix. In the same perspective, our method deals with the auto-correlation of $F^1$ (and $F^2$); specifically, it enforces them to be the same (i.e., $W_p^\top W_p$). This does not mean that our method decorrelates features unless $W_p^\top W_p$ is a diagonal matrix. However, similar to BYOL as discussed in Sec. 3.3, $W_p$ approaches a diagonal matrix with training iteration $t$ in our method as well, although its effect is weaker than BYOL and others; see Fig. 1.

When $W_p = I$, this effectively eliminates the predictor, resulting in our method getting closer to VICReg. The remaining differences are as follows: hinged loss vs. $\ell_2$ norm for the correlation constraint term, a different treatment of diagonal/non-diagonal elements of the covariance matrix vs. their equal treatment, and the use of auto-covariance vs. auto-correlation.

As discussed in [16], there is an explanation based on information theory as to why decorrelating features leads to learning a good representation. However, it is not a rigorous proof since the losses of Barlow-Twins and VICReg differ from the ideal objective function suggested by the theory. The same is true of our method. The implication of the goodness of feature decorrelation may be effective on our method as well.

### 4.4 Partitioned Correlation Constraint Term

We found through preliminary experiments that partitioning the correlation constraint term yields better results. To be specific, we partition $F^1$, the set of features from a single batch, into $k$ subsets of an equal size $n' = n/k$, as $F^1 = (1/\sqrt{k})[F_1^1, F_2^1, ..., F_k^1]$. We then modify the loss (17) as

$$\mathcal{L} = c_{\text{inv}}\|W_p F^1 - F^2\|_{\text{F}}^2 + \frac{c_{\text{cov}}}{d}\sum_{i=1}^{k}\|W_p^\top W_p - W_p(F_i^1 F_i^{1\top})W_p^\top\|_{\text{F}}^2. \qquad (19)$$

We will experimentally show the effectiveness of this partitioned loss later. Its theoretical validation will be left to future research. A small remark is that the partitioned loss is suitable for distributed training of the model in a data-parallel manner. We need only to compute each partitioned term separately on a single computational node, e.g., a GPU. Note that this method differs from those found to be effective in [15]. Specifically, we do *not* introduce different variances for each partition, while theirs use different variances for multiple branches of the Siamese networks. Our method synchronizes the mean and variance among all the nodes, unlike their asymmetrically synchronized "AsymBN." See the pseudo-code shown in the supplementary material for more details.

## 5 Experimental Results

### 5.1 Experimental Setting

**Architecture** We follow the previous studies for the network architecture. Specifically, we use Resnet-50 [11] without the last fully-connected layer (i.e., the classification layer) as a backbone. Following [1, 16], we zero-initialize the weights of the last batch normalization layer in each residual branch (i.e. zero_init_residual=True in PyTorch). We then employ a projector on top of the backbone, which is a MLP with two or three layers basically having the same width, each intermediate layer

having BatchNorm and ReLU. Following the previous methods but BYOL, we synchronize all the batch norm layers, including the standardization layer used for $F^1$ and $F^2$, across different devices, e.g., using SyncBatchNorm of Pytorch. Our method uses a predictor as with BYOL/SimSiam. Note however that ours is a linear predictor, in contrast with BYOL/SimSiam whose default settings employ non-linear predictor, i.e., a two-layer MLP with ReLU at the intermediate layer.

**Devices**   We use a machine equipped with two Intel Xeon Platinum 8360Y Processors, eight NVIDIA A100 GPU and $520$ GiB DDR4 RAM.

**Augmentation**   We follow the procedure of BYOL [10] and Barlow-Twins [16] for data augmentation creating two views of input images. Specifically, we build an augmentation pipeline in the following order: random cropping followed by resizing to $224 \times 224$, horizontal flipping, color jittering, grayscale convertion, Gaussian blurring, and solarization. While random cropping with resizing is always applied, others are probabilistically applied with randomly chosen parameters. Following the above papers, the application of blurring and solarization is asymmetric, i.e., their probabilities differ between the two views. SimSiam and VICReg employ symmetric setting.

**Optimization**   We also follow BYOL and Barlow-Twins for the optimizer and its hyperparameter settings. To be specific, LARS is used along with cosine learning rate decay with 10-epoch linear warm-up; weights in BatchNorm and biases are excluded from weight decay and LARS adaptation. We set the base learning rate $= 0.3$ for the whole model, which is multiplied by BatchSize$/256$. We train our model for 100 epochs, and the employed learning rate decay to $0.$ at the 100-th epoch. We set the batch size to $2048$.

**Evaluation**   For evaluation of methods, we follow the previous studies [1, 6, 10, 16]. Specifically, we use ImageNet (ILSVRC2012) [7] for all the experiments. We use the proposed self-supervised learning method to train the above backbone, along with a projector and a predictor, using all the images of the training split, where the above augmentation and optimization procedures are employed. We then evaluate the method's performance by training and testing a linear classifier using the features extracted by the backbone. We follow the standard evaluation protocol employed in the previous studies. Specifically, we train a linear classifier with SGD with momentum $= 0.9$, batch size $= 256$, weight decay $= 10^{-6}$, and learning rate subject to cosine scheduling with the base rate $= 0.3$, for 100 epochs.

## 5.2   Feature Decorrelation

The previous sections show that the asymmetry tricks implicitly enforce feature decorrelation. To experimentally validate this, we analyze how the auto-correlation matrix $\Sigma$ of extracted features (i.e., $f_i^1$'s) changes during training. For input images, we use all the images of the ImageNet validation split without applying random data augmentation. Figure 1 shows how $\|\Sigma - I\|^2$ changes during the training of a network by different methods. We can see that all the tested methods decrease $\|\Sigma - I\|^2$ with epochs, including those using the asymmetry tricks (i.e., BYOL and SimSiam) and those using different forms of explicit decorrelation constraint (i.e., Barlow-Twins, VICReg, and ours). It is noteworthy that this applies to the BYOL with a linear predictor (i.e., the assumed configuration) and also BYOL/SimSiam with a non-linear predictor.

## 5.3   Performance with Different Configurations

Our method has three hyperparameters, i.e., the partition number $k$, the weight $c_{\mathrm{inv}}$ of the invariance term, and the weight $c_{\mathrm{cov}}$ of the correlation term, as in (19). We conduct experiments to examine how their choice affects the performance using the above evaluation procedure (i.e., ImageNet and linear probe evaluation). We use a projector whose size is 2048-2048-2048 here. Table 1 shows the results for several combinations of these parameters. We can observe that partitioning the correlation loss and setting the weights as $c_{\mathrm{inv}} > c_{\mathrm{cov}}$ yield better performance.

The decorrelation-based methods, such as Barlow-Twins and VICReg, are reported to show better performance with a larger projector. We examine our method's dependency on the size of the projector. Table 2 shows the results. It is observed that a larger projector leads to better performance, similar to Barlow-Twins/VICReg; this tendency persists for different hyperparameter settings.

Table 3: **Effects of feature standardization and an identity predictor.** STD and $I$-PRED indicate the use of the standardization and an identity predictor, respectively.

| INV | COV | STD | $I$-PRED | STOPGRAD | ACC |
|-----|-----|-----|----------|----------|-----|
| 1 | 0 | | | | COLLAPSE |
| 1 | 1 | | | | COLLAPSE |
| 1 | 0 | ✓ | | | 50.9 |
| 1 | 1 | ✓ | | | 61.4 |
| 1 | 0 | | ✓ | | COLLAPSE |
| 1 | 1 | | ✓ | | 62.6 |
| 1 | 0 | ✓ | ✓ | | 50.9 |
| 1 | 1 | ✓ | ✓ | | 58.7 |
| 1 | 0 | ✓ | ✓ | ✓ | 55.0 |

## 5.4 Feature Standardization, Predictor, and StopGrad

As explained in Sec. 4.2, we apply standardization to the features, intending to prevent collapse. We conduct experiments to verify its effects.

As shown in Fig. 1, our method shows a similar effect of feature decorrelation to existing methods, which is caused by $W_p$'s behaviour that $W_p W_p^\top$ approaches diagonal. This raises a question what if we set $W_p = I$ from the beginning. Setting $W_p = I$ means that we do not use a predictor, making our method further closer to the decorrelation methods. Thus, we additionally examine this in the experiments.

Table 3 shows the results. (We fixed $k = 1$ and $c_{inv} = 1$ and set $c_{cov} = 0$ or 1 for simplicity. The projector size is set to 2048-2048-2048.) First, we can confirm that the feature standardization helps prevent collapse independently of the presence of the correlation term (i.e., $c_{cov} = 0$ or 1).

Second, we can see that fixing $W_p = I$, or equivalently eliminating the predictor, works well at least with this specific configuration. However, our method having a learnable predictor with a different configuration achieves the best performance, as will be shown later.

Third, "StopGrad" in the table indicates the case of applying StopGrad similar to BYOL/SimSiam. Comparing the result with/without it under the same configuration (i.e., $c_{inv} = 1$ and $c_{cov} = 0$), we can see StopGrad improves the performance, i.e., $50.9\% \rightarrow 55.0\%$. Switching StopGrad with our correlation term yields further better results, i.e., $55.0\% \rightarrow 61.4\%$.

Finally, we can see that simply removing StopGrad from BYOL/SimSiam will lead to collapse. We need an additional method to prevent collapsing, i.e., either a standardization layer or COV + $I$-PRED (i.e., the correlation constraint term plus the elimination of the predictor).

## 5.5 Comparison with Previous Methods

Table 4 compares the best performance of our method with other methods[1]. As methods' performance varies depending on the size of projectors, we compare them for a fixed projector size. With the projector size of 2048-2048-2048, our method achieves 67.3%, which is better than Barlow-Twins (63.4%) and VICReg (65.1%) and slightly lower than SimSiam (68.1%); SimSiam's higher performance may be attributable to the use of a non-linear predictor. With 8192-8192-8192, our method outperforms Barlow-Twins and VICReg again, i.e., 69.0% vs. 68.6%. BYOL tends to learn slower due to the use of a momentum encoder and performs worse for epochs $= 100$.

## 6 Conclusion and Discussion

In this paper, we first theoretically showed (under some assumptions) that minimizing the loss (4) with StopGrad and a predictor (i.e., BYOL/SimSiam) leads to that (a) $W_p^\top W_p \rightarrow F^1 F^{1\top}$ as $t \rightarrow \infty$. We then showed an empirical observation with BYOL (and many others including ours) that (b) $W_p^\top W_p \rightarrow$ diagonal with increasing $t$. We proposed a novel method incorporating an explicit regularizer to enforce (a) while removing StopGrad. Finally, we presented experimental results

---

[1] We used the official code for these methods other than VICReg.

Table 4: **Results of linear probe evaluation on ImageNet.** We run all methods for 100 training epochs before the training of a linear classifier. Note that only for VICReg, we copy-paste the results in the same setting from [1].

| Methods | Projector | Predictor | Acc |
|---|---|---|---|
| BYOL | 4096-256 | 4096-256 | 66.7 |
| SimSiam | 2048-2048-2048 | 512-2048 | 68.1 |
| Barlow-Twins | 8192-8192-8192 | - | 68.6 |
| | 2048-2048-2048 | - | 63.4 |
| VICReg | 8192-8192-8192 | - | 68.6 |
| | 2048-2048-2048 | - | 65.1 |
| Ours | 4096-256 | 256 | 63.7 |
| | 2048-2048-2048 | 2048 | 67.3 |
| | 8192-8192-8192 | 8192 | 69.0 |

that the method works; it learns as good representation as BYOL/SimSiam even though it does not use StopGrad. We have also presented some extensions (Sec. 4.2 and 4.4) that empirically help convergence and better learning. These all point to the connection between BYOL/SimSiam and the decorrelation-based methods.

**Acknowledgements**

This work was supported by JST [Moonshot Research and Development], Grant Number [JP-MJMS2032] and by JSPS KAKENHI Grant Number 20H05952 and 19H01110.

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
