# Bridging the Gap from Asymmetry Tricks to Decorrelation Principles in Non-contrastive Self-supervised Learning

## A Proof of Lemma 3.2

*Proof of **Lemma** 3.2.* We can expand the loss (4), i.e.,

$$\mathcal{L} = \frac{1}{2}\mathbb{E}_x[\|W_\mathrm{p}f^1 - \mathrm{StopGrad}(f^2)\|^2]$$

as follows:

$$
\begin{aligned}
\mathcal{L} &= \frac{1}{2}\mathbb{E}_x[\mathrm{tr}(W_\mathrm{p}f^1 f^{1\top}W_\mathrm{p}^\top) - 2\mathrm{tr}(\mathrm{StopGrad}(f^2)f^{1\top}W_\mathrm{p}^\top) + \mathrm{tr}(\mathrm{StopGrad}(f^2 f^{2\top}))] \\
&= \frac{1}{2}(\mathrm{tr}(W_\mathrm{p}\mathbb{E}_x[f^1 f^{1\top}]W_\mathrm{p}^\top) - 2\mathrm{tr}(\mathbb{E}_x[\mathrm{StopGrad}(f^2)f^{1\top}]W_\mathrm{p}^\top) + \mathrm{tr}(\mathbb{E}_x[\mathrm{StopGrad}(f^2 f^{2\top})])) \\
&= \frac{1}{2}(\mathrm{tr}(W_\mathrm{p}F^1 F^{1\top}W_\mathrm{p}^\top) - 2\mathrm{tr}(\mathrm{StopGrad}(F^2)F^{1\top}W_\mathrm{p}^\top) + \mathrm{tr}(\mathrm{StopGrad}(F^2 F^{2\top})])).
\end{aligned}
\tag{20}
$$

We used $\mathbb{E}_x[f^1 f^{1\top}] = F^1 F^{1\top}$, $\mathbb{E}_x[f^2 f^{1\top}] = F^2 F^{1\top}$, and $\mathbb{E}_x[f^2 f^{2\top}] = F^2 F^{2\top}$.

Taking derivatives of $\mathcal{L}$ with respect to $W_\mathrm{p}$ yields (7) since

$$\frac{\partial \mathcal{L}}{\partial W_\mathrm{p}} = \frac{1}{2}(2W_\mathrm{p}F^1 F^{1\top}) - F^2 F^{1\top} = W_\mathrm{p}F^1 F^{1\top} - F^2 F^{1\top}. \tag{21}$$

Note that there is no gradient at $\mathrm{StopGrad}(\cdot)$. Similarly, the derivative with respect to $F^{1\prime}$ is given by (8) since

$$\frac{\partial \mathcal{L}}{\partial F^1} = \frac{1}{2}(2W_\mathrm{p}^\top W_\mathrm{p}F^1) - W_\mathrm{p}^\top F^2 = W_\mathrm{p}^\top W_\mathrm{p}F^1 - W_\mathrm{p}^\top F^2. \tag{22}$$

$\square$

## B Proof of Lemma 3.3

*Proof of **Lemma** 3.3.* Assume the mapping from the input $x \in \mathbb{R}^P$ to $f \in \mathbb{R}^D$ is given by a linear transformation $f = Wx$. (Or maybe affine $f = Wx + b$.) We will use a 'vectorized' representation of $W = [w_1, \ldots, w_D]^\top$ as $w = [w_1^\top, \ldots, w_D^\top]^\top \in \mathbb{R}^{DP}$. Now, suppose we change $w$ as $w \to w + \delta w$, where we choose the gradient of $L$ for $\delta w$ as

$$\delta w = -\frac{\partial L}{\partial w}. \tag{23}$$

Then, $f$ will change as $f \to f + \delta f$ accordingly, where

$$\delta f = \frac{\partial f}{\partial w}\delta w = -\frac{\partial f}{\partial w}\frac{\partial L}{\partial w}. \tag{24}$$

Since $w$ affects the loss $L$ only through $f$, we use chain rule to get

$$\frac{\partial L}{\partial w} = \left(\frac{\partial f}{\partial w}\right)^\top \frac{\partial L}{\partial f}. \tag{25}$$

16  Thus, substituting this into (24) yields

$$\delta f = -\frac{\partial f}{\partial w}\left(\frac{\partial f}{\partial w}\right)^{\top}\frac{\partial L}{\partial f}. \tag{26}$$

17  Since $f = [w_1^{\top}x, \ldots, w_D^{\top}x]^{\top}$,

$$\frac{\partial f}{\partial w} = \begin{bmatrix} x^{\top} & & & \\ & x^{\top} & & \\ & & \ddots & \\ & & & x^{\top} \end{bmatrix} \ (\in \mathbb{R}^{D \times DP}) \tag{27}$$

18  Thus,

$$\frac{\partial f}{\partial w}\left(\frac{\partial f}{\partial w}\right)^{\top} = \begin{bmatrix} x^{\top}x & & & \\ & x^{\top}x & & \\ & & \ddots & \\ & & & x^{\top}x \end{bmatrix} = (x^{\top}x)I \tag{28}$$

19  Substituting this into (26) yields

$$\delta f = -\frac{\partial f}{\partial w}\left(\frac{\partial f}{\partial w}\right)^{\top}\frac{\partial L}{\partial f} = -(x^{\top}x)\frac{\partial L}{\partial f}. \tag{29}$$

20  This states that when we move $w$ in the direction $\delta w$ of minimizing $L$, $f$ moves in the direction of
21  $-\partial L/\partial f$.

22  Next, we consider the effect of weight decay on $w$. Then $\delta w$ becomes $-\partial L/\partial w - \eta w$. We consider
23  the change $\delta f'$ of $f$ due to $-\eta w$. This is given by

$$\delta f' = \frac{\partial f}{\partial w}(-\eta w) \tag{30}$$

24  Using (27),

$$\delta f' = -\eta \begin{bmatrix} x^{\top}w_1 \\ x^{\top}w_2 \\ \vdots \\ x^{\top}w_D \end{bmatrix} = -\eta W x = -\eta f \tag{31}$$

25  In conclusion, when we move $w$ as $w \to w - \partial L/\partial w - \eta w$, $f$'s change $\delta f$ will be given by

$$\delta f = -(x^{\top}x)\frac{\partial L}{\partial f} - \eta f. \tag{32}$$

26  Supposing $x$ to be an ImageNet image, we may think $x^{\top}x \sim \text{const}$. Assuming $x^{\top}x = 1$, the above
27  leads to $\dot{F} = -\partial L/\partial F - \eta F$, which is (11). $\qquad\square$

## C  Behavior of $W_{\mathrm{p}}$ in Ours

29  We also examine how $W_{\mathrm{p}}$ changes in the optimization of our proposed method. Following the same
30  experimental setting as Sec. 5, we train a linear predictor with $W_{\mathrm{p}} \in \mathbb{R}^{8192 \times 8192}$. As shown in Fig. 3,
31  $W_p$ approaches to a diagonal matrix during training process.

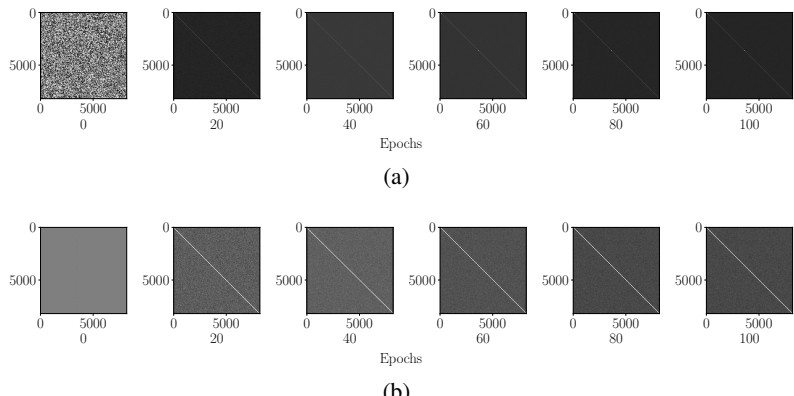

(a)

(b)

Figure 3: (a) $W_p$ in our proposed method at different epochs of training time. (b) Normalized $W_p$ for better visualization; each row of $W_p$ is divided by its diagonal entry.

## D  More Results with a Broader Range of Configuration

Table 5 shows more results of our method with different configurations and hyperparameter settings.

Table 5: Results of our method on more configurations.

| PROJECTOR | PREDICTOR | LR | #PARTITION | INV COEF | COV COEF | ACC@1 |
|---|---|---|---|---|---|---|
| 4096-256 | 256 | 0.3 | 1 | 1 | 1 | 54.7 |
| 4096-256 | 256 | 0.3 | 1 | 10 | 1 | 56.8 |
| 4096-256 | 256 | 0.3 | 1 | 1 | 10 | 53.1 |
| 4096-256 | 256 | 0.3 | 8 | 10 | 1 | 63.8 |
| 4096-256 | 256 | 0.3 | 32 | 1 | 1 | 54.8 |
| 4096-256 | 256 | 0.3 | 32 | 10 | 1 | 63.7 |
| 2048-2048-2048 | 2048 | 0.3 | 1 | 1 | 0 | 50.9 |
| 2048-2048-2048 | 2048 | 0.3 | 1 | 1 | 1 | 61.4 |
| 2048-2048-2048 | 2048 | 0.3 | 1 | 1 | 25 | 60.6 |
| 2048-2048-2048 | 2048 | 0.3 | 1 | 25 | 1 | 60.1 |
| 2048-2048-2048 | 2048 | 0.3 | 1 | 25 | 25 | 63.4 |
| 2048-2048-2048 | 2048 | 0.3 | 8 | 25 | 1 | 67.3 |
| 2048-2048-2048 | 2048 | 0.3 | 8 | 25 | 25 | 66.2 |
| 2048-2048-2048 | 2048 | 0.3 | 32 | 10 | 1 | 66.6 |
| 2048-2048-2048 | 2048 | 0.3 | 32 | 25 | 1 | 67.3 |
| 2048-2048-2048 | 2048 | 0.3 | 32 | 50 | 1 | 67.2 |
| 2048-2048-2048 | 2048 | 0.45 | 1 | 1 | 1 | 63.9 |
| 2048-2048-2048 | 2048 | 0.45 | 1 | 25 | 25 | 64.7 |
| 2048-2048-2048 | 2048 | 0.45 | 32 | 10 | 1 | 64.3 |
| 8192-8192-8192 | 8192 | 0.3 | 8 | 25 | 8 | 69.0 |

## E  Standardization layer

Feature standardization is computed as following, independent to partitions:

$$Std(f_i) = \frac{f_i - \mu_{i,sync}}{\sigma_{i,sync}} \tag{33}$$

,where $f_i$ is the $i$th entry of $f$, $\mu_{i,sync}$ and $\sigma_{i,sync}$ are synchronized $i$th entry of mean and deviation respectively among devices.

 # F Pseudo-code of our proposed method

---

**Algorithm 1** Our Proposed Method, PyTorch-like

---

```python
# h: backbone + projector
# w: weight of predictor
# D: projector output size
# C_in, C_cov: coefficients
#
# In this pseudo-code, we assume number of partitions equals to
    number of gpus, and the following code is processed on a single
    gpu.
for x in loader:
    x1, x2 = aug1(x), aug2(x)
    f1, f2 = h(x1), h(x2)
    f1, f2 = std(f1), std(f2)

    # Processed by predictor
    p1, p2 = f1 @ w.T, f2 @ w.T
    inv_loss = (p1-f2).pow(2).mean() + (p2-f1).pow(2).mean()

    # Note: we do not collect cov from
    # different gpus
    wtw = w.T @ w
    n = p1.size(0) # Batch size per gpu
    cov1 = p1.T @ p1 / n
    cov2 = p2.T @ p2 / n
    cov_loss = (cov1-wtw).pow(2).sum() + (cov2-wtw).pow(2).sum()

    loss = C_in * inv_loss + C_cov / D * cov_loss
    loss.backward()
    update(f,w)

def std(f): # Standarization
    return SyncBN(affine=False)(f)
```

---