# OpenReview forum: "Bridging the Gap from Asymmetry Tricks to Decorrelation Principles in Non-contrastive Self-supervised Learning"
_NeurIPS.cc/2022/Conference — NeurIPS 2022 Accept_

### Official Review · Reviewer_LRxj · 2022-07-04

**Rating:** 5
**Confidence:** 5
**Soundness:** 2 fair
**Presentation:** 3 good
**Contribution:** 2 fair

**Summary:**

This paper studies the dynamics behind non-contrastive self-supervised learning methods. It shows that the stop-grad approach (e.g., BYOL, SimSiam) is effectively also doing feature decorrelation, similar to BarlowTwins, VICReg. The authors start with Tian et al. (arXiv:2102.06810)'s results and show that the stop-grad will satisfy W(t)^TW(t) = F(t)F(t)^T at large step t. The authors propose a novel method that explicitly uses this term as a regularizer. This can be seen as a soft version of BarlowTwins that require decorrelation but is not necessarily full rank. Experiment results (100ep ImageNet pretraining, linear probe) show that the proposed method outperforms BarlowTwins and VICReg.

**Questions:**

1. What's the consideration for using a different projector in BarlowTwins and VICReg? The hyperparameters in these methods are optimized for the original projector size. In fact, the accuracy could be much higher if a different set of hyperparameters were used for such a projector head. I   suggest not to claim the advantage of a proposed method by beating a weakened baseline.

2. What's the consideration of "partition"? This is not discussed in the theory part or the motivation. How to understand this term?

**Strengths And Weaknesses:**

Strengths:
1. The proposed method is well motivated by the theory, which is an extension of Tian's (arXiv:2102.06810) results.
2. The ablation study is convincing. It proves the effectiveness of the proposed regularizer term.
3. The authors provided all experimental details, including pseudo-code and hyperparameters tuning.

Weaknesses
1. The authors claim that they extend the theory. However, most of the equations are already known in the literature. For example, any linear system under SGD will satisfy eq14; see arXiv:1905.13655, arXiv:2110.09348, arXiv:2201.12680. Therefore, I don't find Section 3.2 supports the claim that "we theoretically show that the use of stop-gradient and a projector implicitly provides a constraint leading to similar feature decorrelation."
2. The primary connection between the theory and the proposed method is that W^TW becomes a diagonal matrix and will not collapse to a low rank. The authors claim that they empirically observe this happens for BYOL, as shown in supplementary material section C. However, there is no study on the dynamics of this term in the proposed method. The authors need to show that this term follows similar behavior.
3. The authors did an extensive study on the projector design. With various projector/predictor dimension settings, the proposed method demonstrates superior performances over BarlowTwins and VICReg. However, this study is less significant considering the main claim. The projector design is hyperparameter tuning. It would be better to provide a clear comparison between optimal settings.

---

> ### Author Response · Authors · 2022-08-02
> **Author response to Reviewer LRxj**
>
> **Q1. The authors claim that they extend the theory. However, most of the equations are already known in the literature. For example, any linear system under SGD will satisfy eq14; see arXiv:1905.13655, arXiv:2110.09348, arXiv:2201.12680. Therefore, I don't find Section 3.2 supports the claim that "we theoretically show that the use of stop-gradient and a projector implicitly provides a constraint leading to similar feature decorrelation.**
>
> We would be grateful if the reviewer shows which parts in the listed papers state `most of the equations,' as we could not find a similar discussion or result in these papers.
>
>
> **Q2. The primary connection between the theory and the proposed method is that $W_p^\top W_p$ becomes a diagonal matrix and will not collapse to a low rank. The authors claim that they empirically observe this happens for BYOL, as shown in supplementary material section C. However, there is no study on the dynamics of this term in the proposed method. The authors need to show that this term follows similar behavior.**
>
> We updated a visualization of predictor's weight $W_p$ for our proposed method in supplementary material section F. We can see $W_p$ asymptotically approaches a diagonal matrix. (We omitted since it was obvious for us.)
>
>
> **Q3. The authors did an extensive study on the projector design. With various projector/predictor dimension settings, the proposed method demonstrates superior performances over BarlowTwins and VICReg. However, this study is less significant considering the main claim. The projector design is hyperparameter tuning. It would be better to provide a clear comparison between optimal settings.**
>
> See the answer to Q4.
>
>
> **Q4. What's the consideration for using a different projector in BarlowTwins and VICReg? The hyperparameters in these methods are optimized for the original projector size. In fact, the accuracy could be much higher if a different set of hyperparameters were used for such a projector head. I suggest not to claim the advantage of a proposed method by beating a weakened baseline.**
>
> The design of the projectors may be merely hyper-parameters for those interested only in the performance of SSL methods. However, it is not hyper-parameters selection for researchers who want to understand the mechanism of these methods. That is why we show the different combinations of methods and projector designs, which may be helpful for some readers. We do not (intend to) compare our method with weakened baselines.
>
>
> **Q5. What's the consideration of "partition"? This is not discussed in the theory part or the motivation. How to understand this term?**
>
> We position the partition method as an option to improve the performance. The proposed method works without it; thus, our argument about the connection between BYOL/SimSiam and BT/VICReg stands without it. We report its results since we found that it improves the performance of the downstream task, which will be helpful for readers.

---

### Official Review · Reviewer_R4L1 · 2022-07-12

**Rating:** 5
**Confidence:** 4
**Soundness:** 3 good
**Presentation:** 3 good
**Contribution:** 3 good

**Summary:**

This paper studies the role of asymmetric “tricks” (like predictor layer and stop-gradient) to prevent feature collapse in *non-contrastive methods* like BYOL/SimSiam. Extending prior theoretical results on a linear representation model [Tian et al., 2021], the paper argues that these tricks implicitly enforce “decorrelation” of features of different inputs, similar to other non-contrastive methods like Barlow-Twins and VICRig, thus establishing a connection between these two paradigms. Based on this analysis, the paper also proposes a non-contrastive method that uses a linear prediction head, explicitly enforces the decorrelation-like constraint that is derived, and does not use stop-gradient. Experiments suggest that the proposed method performs similarly (or better) than all of the aforementioned non-contrastive method on ImageNet linear probe evaluation.

**Questions:**

Other minor points

- Missing references: [1, 2, 3] either theoretically or empirically study non-contrastive learning (although [3] is very recent). [4] theoretically studies non-contrastive method (SimSiam) with feature normalization and connects it to non-linear CCA, providing a justification for feature decorrelation beyond an information theoretic one.
- Theorem 3.1 looks similar to Theorem 1 from reference [14] from the paper. It would help to compare and discuss the differences from it. The proof of Theorem 3.1, especially the jumps from eq. 11-13 can be written more clearly. Also useful to statement the assumption that F is linear in the input in the theorem and lemma statements.
- In the setting from Theorem 3.1, what happens in the dynamics if there is no stop-gradient? Will $W$ not converge to $I$ and there will be no feature correlation?
- L170-171: Editing issue


References

[1] X. Wang, X. Chen, S. Du, Y.Tian, Towards Demystifying Representation Learning with Non-contrastive Self-supervision. 2021

[2] X. Wang, H. Fan, Y. Tian, D. Kihara, X. Chen. On the Importance of Asymmetry for Siamese Representation Learning. CVPR 2022

[3] Z. Wen, Y. Li. The Mechanism of Prediction Head in Non-contrastive Self-supervised Learning. 2022

[4] J. Lee, Q. Lei, N. Saunshi, J. Zhou. Predicting What You Already Know Helps: Provable Self-Supervised Learning. NeurIPS 2021

**Limitations:**

Some limitations are discussed. No societal impact discussed; I do not forsee any immediate negative impacts.

**Strengths And Weaknesses:**

**Strengths**

(S1) The paper is well motivated and studies a relevant question about why non-contrastive methods perform well and avoid trivial solutions that collapsing the representations

(S2) Aims to bridge two sets of ideas to avoid collapse: asymmetric tricks and feature decorrelation

(S3) Paper is clearly written and easy to follow for the most part

(S4) Extensive experiments are conducted and useful ablation studies are reported. The rough conclusion is that the proposed regularization term (along with the standardization) is competitive with previous non-contrastive methods and is a better alternative to stop-gradient.




**Weaknesses**

My main concern with the paper is about the connetion to feature decorrelation in section 3.3 and the modification to the proposed regularization in eq 17, both of which are crucial points.

(W1) Section 3.3 is the main (probably only) part that supports the paper’s claim about connection between asymmetric tricks and feature decorrelation. However I found this section to be “hand-wavy” and imprecise. To elaborate, the theoretical result suggests that feature correlation for stop-gradient dynamics will converge to $WW^{\top}$ and if the predictor $W$ converges to $I$, then feature correlation will also converge to $I$ as desired. While the paper makes an interesting empirical observation that $W$ converges to $I$ in practice for many methods, the theoretical justification seems weak as currently presented. It suggests that explanation for $W \rightarrow I$ follows from Tian et al.’s results, but it is not clear how exactly it follows.
- L139: “each eigenvalue of $W_p$ will be either 0 or a  positive value $\lambda^*$ at its stable convergence/equilibrium state”. Why is this true? Which result of Tian et al. suggests this?
- L143: “If some of its eigenvalues are 0, $W_p$ is close to a diagonal matrix”. This is not necessarily true. If some eigenvalues are 0 then $W_p$ will be low rank and how close to diagonal it is will depend on the span of the non-zero eigenvectors of $W_p$.

Overall I find the connection to feature decorrelation unconvincing until more explanation is provided. What helps the discussion is the empirical finding that $W$ converges to identity.




(W2)  Modification to regularization: The modification from $|W^{\top} W - F F^{\top}|^2$ to $|W^{\top} W - W F F^{\top} W^{\top}|^2$ seems benign, but it is not clear how benign it is. Although the objective does not change a lot with this when the first and second terms are sufficiently small, since non-contrastive is all about the dynamics and not the objective, this could potentially play a big role. Furthermore if $W$ is assumed to be symmetric, then $|W^{\top} W - W F F^{\top} W|^2 = |W (I - F F^{\top}) W|^2$ which almost directly enforces feature decorrelation for a full rank $W$. It would be good to get a clarification about this point from the authors and it might also help to report experiment results when using the original regularization $|W^{\top} W - F F^{\top}|^2$.



Given these issues, I am currently leaning towards a **weak reject**. Some other minor issues and questions are listed below.

---

> ### Author Response · Authors · 2022-08-02
> **Author response to Reviewer R4L1 (Q1-Q2)**
>
> **Q1. Section 3.3 is the main (probably only) part that supports the paper’s claim about connection between asymmetric tricks and feature decorrelation. However I found this section to be “hand-wavy” and imprecise. ...... It suggests that explanation for $W_p\rightarrow I$ follows from Tian et al.’s results, but it is not clear how exactly it follows.**
>
> A brief summary of our argument is as follows. We present a theoretical proof of $W_p^\top W_p\rightarrow F^1 F^{1\top}$ and an empirical observation of $W_p^\top W_p\rightarrow \mathrm{diagonal}$. The combination of these two implies the connection from BYOL/SimSiam to decorrelation-based methods. The experimental results that the derived new non-contrastive method successfully learns good representation reinforces the implication.
>
> To present the above argument more clearly in our manuscript, we have rewritten Sec 3.3; it now focuses more on the empirical evidence of $W_p^\top W_p\rightarrow \mathrm{diagonal}$. (It also explain the implication of its combination with the theoretical result, i.e., $W_p^\top W_p\rightarrow F^1 F^{1\top}$.) Although the current version says ``details are provided in the supplementary'' due to the space limitation, we will move them to Sec.~3.3 of the final version of the main paper, as it will have more space.
>
> We have removed the parts in question in Q1-1 and Q1-2 from Sec. 3.3. Thus, we explain below what we intended to say in the removed parts.
>
>
> **Q1-1. "each eigenvalue of $W_p$ will be either 0 or a positive value $\lambda^\*$ at its stable convergence/equilibrium state". Why is this true? Which result of Tian et al. suggests this?**
> A precise derivation is as follows. Following (Tian et al. 2020) and particularly their Theorem 3, we can discuss the dynamics of $W_p$ and $F$ sorely with their eigenvalues $p_j$ and $s_j$, respectively; see (11) and (12) in (Tian et al. 2020). Since we do not use a momentum encoder, these are simplified as
>
> \begin{equation}
>     \dot p_j = s_j[1-(1+\sigma^2)p_j] -\eta p_j       \tag{35}
> \end{equation}
>
> \begin{equation}
>     \dot s_j = 2p_j[1-(1+\sigma^2)p_j] - 2\eta s_j.   \tag{36}
> \end{equation}
>
> Combining these two, we have
>
> \begin{equation}
>     s_j(t) = p^2_j(t)+c_je^{-2\eta t}, \tag{37}
> \end{equation}
>
>
>
>
> where $c_j$'s are the initial conditions. When $t\rightarrow\infty$, $s_j(t)\approx p^2_j(t)$. Substituting $s_j(t)\approx p^2_j(t)$ back into Eq (35) leads to $\dot p_j = p^2_j(t)[1-(1+\sigma^2)p_j] -\eta p_j$. There are three solutions for this equation, which are $p_j^{*\pm}=\frac{1\pm\sqrt{1-4\eta(1+\sigma^2)}}{2(1+\sigma^2)} $ and $0$. As shown in Fig. 4 of (Tian et al. 2020), only $0$ and $p_j^{\*+}$ are stable solutions. The derivation is valid for all the eigenvalues $p_j$'s, and thus we can say that $p_j$ are either $0$ or $p^{\*+}_{j}$ for any $j$ as $t\rightarrow\infty$.
>
>
> **Q1-2. “If some of its eigenvalues are 0, $W_p$ is close to a diagonal matrix”. This is not necessarily true. If some eigenvalues are 0 then $W_p$ will be low rank and how close to diagonal it is will depend on the span of the non-zero eigenvectors of $W_p$.**
>
> The reviewer is correct. Our original statement relies on another empirical observation shown below. Thus, we have eliminated the above argument from Sec. 3.3. We conducted a simplified simulation in which we randomly generated a matrix $W=VDV^\top$; $V$ is a random orthogonal matrix, and $D$ is a diagonal matrix whose diagonal entries are either 1 or 0. We can observe from the results that $W$ approaches a diagonal matrix when $p$ increases or its size (i.e., feature dimension) increases. This indicates that even when $W_p$ has a certain amount of zero eigenvalues, as long as they are not too many, $W_p$ approaches a diagonal matrix if its size is large.
>
>
> **Q2. Modification to regularization: The modification from $\|W_p^\top W_p-FF^\top\|^2$ to $\|W_p^\top W_p-W_pFF^\top W_p^\top\|$ seems benign, but it is not clear how benign it is. ........ It would be good to get a clarification about this point from the authors and it might also help to report experiment results when using the original regularization ($\|W_p^\top W_p-FF^\top\|^2$).**
>
> While we omitted in our initial manuscript, the regularizer $\|W_p^\top W_p-FF^\top\|^2$ in (16) actually works; see the table in our reply to Reviewer for the comparison with $\|W_p^\top W_p-W_pFF^\top W_p^\top\|$ in (17) in the same setting of the experiments of Sec. 5. We introduce $\|W_p^\top W_p-W_pFF^\top W_p^\top\|$ since the original one tends to be more sensitive to the choice of the regularization weight (i.e., $c_\mathrm{cov}$ in (17)); there is no more reason.
>
>
> | Loss | Projector      | #Part | INV | COV  | ACC  |
> |------|----------------|-------|-----|------|------|
> | (16) | 8192-8192-8192 | 8     | 25  | 0.96 | 68.3 |
> | (17) | 8192-8192-8192 | 8     | 25  | 8    | 69   |

---

> ### Author Response · Authors · 2022-08-02
> **Author response to Reviewer R4L1 (Q3-Q5)**
>
> **Q3. Missing references: [1, 2, 3] either theoretically or empirically study non-contrastive learning (although [3] is very recent). [4] theoretically studies non-contrastive method (SimSiam) with feature normalization and connects it to non-linear CCA, providing a justification for feature decorrelation beyond an information theoretic one.**
>
> We will add the following texts in Sec.~2 of the final version, which explain and compare these studies. We do not include it in the current version due to the space limitation.
>
> Several studies investigate how BYOL/SimSiam can learn good representation successfully. Wang et al. [1] argue that weight decay acts like a threshold for distinguishing two types of features, i.e., features invariant to input augmentations and those with high variance. Wang et al. [2] show empirical evidence that the variance between $f_1$ and $f_2$ plays an essential role in self-supervised learning, whether it is a contrastive or non-contrastive method; a gap in their variances leads to better representation learning. Although they call the architectures introducing a variance gap "asymmetric", it differs from "asymmetric tricks'' in our study. It is unclear how it relates to these two studies [2,3], as ours deals with considerably different problems. Wen et al. [3] analyze BYOL/SimSiam under the assumption of a linear "diagonal predictor," which has trainable parameters only in the diagonal entries. They then categorize training into two phases called substitution and accelerating phases. While it differs from our study, which considers more general linear predictors, their study seems to share something with ours, e.g., the implementation of the standardization layer. (We need more time to investigate the dis(similarity) with ours, as their paper was published very recently.) Finally, Lee et al. examine the relationship between the learned features by SimSiam and the performance of downstream tasks, finding the similarity of SimSiam with canonical correlation analysis (CCA); for instance, SimSiam maximizes $\cos(W_pWx_1, \mathrm{StopGrad}(Wx_2))$ whereas CCA maximizes $(Wx_1)^\top(Wx_2)$. While their study shares a goal similar to ours, it is noteworthy that they introduce a constraint whitening the features, which is not (explicitly) employed by SimSiam; the constraint is indispensable in their analysis, which seems to weaken their argument somewhat.
>
>
> **Q4. Theorem 3.1 looks similar to Theorem 1 from reference Tian et al.[14] It would help to compare and discuss the differences from it. The proof of Theorem 3.1, especially the jumps from eq. 11-13 can be written more clearly. Also useful to statement the assumption that F is linear in the input in the theorem and lemma statements.**
>
> Our Theorem 3.1 differs from Theorem 1 in (Tian et al. 2020) in that we consider the dynamics of $W_p$ and $F^1$, not $W_p$ and $W$ (the projector's weight) in theirs, and we show $W_p^\top W_p=F^1 F^{1\top}$, not $\lVert W_pF^1F^{2\top} - F^1F^{2\top}W_p\rVert\rightarrow 0$ in theirs. We think (11)-(13) is already clear.
>
>
> **Q5. In the setting from Theorem 3.1, what happens in the dynamics if there is no stop-gradient? Will not converge to and there will be no feature correlation?**
>
> If there is no StopGrad, then (9) will not hold, and $F^1$ will have a different dynamics due to the effect of $\frac{\partial \mathcal{L}}{\partial F^2}$.

---

> ### Comment · Reviewer_R4L1 · 2022-08-08
> **Thank you for the response**
>
> I thank the authors for their detailed response and clarifications. I found the response to my two main concerns (Section 3.3 and regularization term) reasonably well addressed. Some quick follow up comments below.
>
> - What happens with no stop-grad? I understand that the current analysis would not work in this case. However failure of analysis does not imply that not having stop-grad is bad for the simple linear setting being analyzed. Some theory/intuitions/citations/experiments to say why stop-grad is *necessary* even in this linear setting will be helpful to convince us that stop-gradient is not independent to the issue of collapsed solutions.
>
> - Section 3.3: The revised version, that only uses the empirical justification for $W$ being diagonal, seems cleaner. However it also seems like the arguments made in Q1-1 and Q1-2 of the author response might also make a compelling story and is worth including in the main paper. Is Q1-1 an exact theorem? If yes, this might also help resolve the issue raised by reviewer LRxj about lack of analysis of dynamics of $W$.
>
> After reading the other reviews that raised some good points and concerns, and the author responses to all the reviews, I am inclined to raise the score to weak accept.

---

> > ### Author Response · Authors · 2022-08-09
> > **Further author response to Reviewer R4L1**
> >
> > We thank Reviewer R4L1's detailed and thoughtful comments. Some further comments from us are listed below.
> > - To the first comment,  fortunately, there is an answer in (Tian et al., 2021); the authors provide theoretical proof that not having stop-grad leads to collapsed solutions under the linear assumption.
> > - To the second question, our answer is yes. What we show in Q1-1 is theoretically true. We derived the result just by simplifying (11) and (12) in (Tian et al., 2021). We agree that Q1-1 and the empirical findings Q1-2 will improve our understanding of the problem. Therefore, if allowed, we will add them to the final version. We will present them as additional remarks to avoid confusion.

---

### Official Review · Reviewer_yFdG · 2022-07-13

**Rating:** 5
**Confidence:** 3
**Soundness:** 3 good
**Presentation:** 3 good
**Contribution:** 3 good

**Summary:**

The paper follows the theoretical framework proposed by Tian, Y. et al in [14], and further shown that the asymmetry in contrastive learning would lead to a feature decorrelation. As a demonstration (and an empirical proposal), the author demonstrated that the stop-gradient in contrastive learning (BYOL/ SimSiam) can be eliminated by explicitly imposing the derived constraint with additional feature standardization. The empirical results majorly supports the claim and theory of the paper. Eventually a better performance is reported with the proposed method on ImageNet (contrastive learning with linear prob).

**Questions:**

Could the author share additional data points on the magnitude comparison between the two losses.

Could the author elaborate what are the bn (or bn without learnable parameters) introduced in the network? Do we have additional bn before the standardization. Do we perform sync for the standardization.






**Limitations:**

The standardization is not introduced/ discussed in the derivative derivation process, which might lead to different conclusion on the decorrelation part. The standardization is only introduced as an empirical trick, which weak the soundness of the paper and make the theory and finding less convincing.

The behavior of partition is completely not explained in the paper. The theory part does not introduce anything would lead to partition, which further weak the soundness of the paper.

The partition behavior (2048 / 32 = 64 per batch) was originally introduced in the paper of "On the Importance of Asymmetry for Siamese Representation Learning", the author did not discuss or cite the paper.

Stop grad or mmt encoder are not the only "asymmetric tricks", there are other asymmetry tricks used in contrastive learning field (for example, multi crop, scalemix, Asym Aug, Mean Encoder and Sync BN, etc. It would make the paper less precise and overclaim if the terminology of "asymmetric tricks" were not well/ better defined.


Minor issues

In the Pseudo-code, the function of standardization was named as bn which can be misleading.

The Pseudo-code is over complicated, it is not necessary to introduce a hyper parameter of GPU. More importantly the num of partition are not necessarily num of gpu.

line 171, the writing is not unfinished.

**Strengths And Weaknesses:**

The paper pinpointed that the a small set of asymmetry tricks (stop gradient, etc) employed by BYOL/SimSiam have an additional implicit effect of feature decorrelation. More importantly, one of the major trick of stop gradient can be removed and replaced with the constraint as an explicit regularizer. This finding provides a better understanding of the working mechanism of the asymmetry tricks.

The formula derivation is sound.

The ablation in the paper is compact and clear.

The paper is in generally fairly elaborated in smooth writing.

The paper reports some meaningful datapoints to supported the claim. In table 3, the ablation prove that the stop grad can be eliminated and replaced with INV + STD (or with I-PRED) to keep the network train without collapse. Consistent datapoints supports the claim that COV helps the network converges better, outperform the counterpart without COV constraint. It is also very convincing to see empirical experiments (with no-augmented image) demonstrated the theory of the correlation constraint.

---

> ### Author Response · Authors · 2022-08-02
> **Author response to Reviewer yFdG**
>
> **Q1. Could the author share additional data points on the magnitude comparison between two losses?**
>
> We will add more data points in the future version as it requires computationally demanding experiments, maybe in the supplementary material due to space limitation.
>
>
> **Q2. Could the author elaborate what are the bn (or bn without learnable parameters) introduced in the network? Do we have additional bn before the standardization. Do we perform sync for the standardization.**
>
> The standardization we use is "bn without learnable parameters," and we do not use an additional bn layer before any standardization. We perform sync and thus the mean and variance are shared among all the devices. We use "std" in the pseudo code to mean this operation. We implement the above using SyncBatchnorm1d with ``affine=False'' in PyTorch. We have added the above to the  supplementary material.
>
>
> **Q3. The standardization is not introduced/ discussed in the derivative derivation process, which might lead to different conclusion on the decorrelation part. The standardization is only introduced as an empirical trick, which weak the soundness of the paper and make the theory and finding less convincing.**
>
> While the effect of standardization is unclear, we think that its necessity is neutral to our main argument since it is independent of the method's category; SimSiam and BarlowTwins need batch normalization whereas BYOL and VICReg do not. The fact that the new method derived from our analysis works (un)expectedly well will be a strong implication that BYOL/SimSiam implicitly performs feature decorrelation similar to BT/VICReg.
>
>
> **Q4. The behavior of partition is completely not explained in the paper. The theory part does not introduce anything would lead to partition, which further weak the soundness of the paper.**
>
> We position the partition method as an option.  Our method (i.e., minimizing (16) or (17)) without it works at the cost of slight performance decrease in the downstream task. The table below compares the results with and without partition in a specific setting of the projector etc. Note \#PARTITION$=1$ is the case without partition. The gap from the best one to the best case with partition is 3.9pp (63.4\% vs. 67.3\%).
>
> | Projector      | Predictor | Lr  | #Part | INV | COV | ACC  |
> |----------------|-----------|-----|-------|-----|-----|------|
> | 2048-2048-2048 | 2048      | 0.3 | 1     | 1   | 0   | 50.9 |
> | 2048-2048-2048 | 2048      | 0.3 | 1     | 1   | 1   | 61.4 |
> | 2048-2048-2048 | 2048      | 0.3 | 1     | 1   | 25  | 60.6 |
> | 2048-2048-2048 | 2048      | 0.3 | 1     | 25  | 1   | 60.1 |
> | 2048-2048-2048 | 2048      | 0.3 | 1     | 25  | 25  | 63.4 |
> | 2048-2048-2048 | 2048      | 0.3 | 8     | 25  | 1   | 67.3 |
> | 2048-2048-2048 | 2048      | 0.3 | 8     | 25  | 25  | 66.2 |
> | 2048-2048-2048 | 2048      | 0.3 | 32    | 10  | 1   | 66.6 |
> | 2048-2048-2048 | 2048      | 0.3 | 32    | 25  | 1   | 67.3 |
> | 2048-2048-2048 | 2048      | 0.3 | 32    | 50  | 1   | 67.2 |
>
> **Q5. The partition behavior (2048 / 32 = 64 per batch) was originally introduced in the paper of "On the Importance of Asymmetry for Siamese Representation Learning", the author did not discuss or cite the paper.**
>
> We first want to clarify that our partition method differs from what they (the authors of the paper) found to be effective. Specifically,
> - We do *not* introduce different variances for each partition, while theirs use different variances for multiple branches of the Siamese networks.
> - Our method synchronized the mean and variance among all the devices, unlike their asymmetrically synchronized "AsymBN."
>
> We have added this explanation about the differences from their study with a proper citation in the updated manuscript to the last of Sec.4.4.
>
>
> **Q6. Stop grad or mmt encoder are not the only "asymmetric tricks", there are other asymmetry tricks used in contrastive learning field (for example, multi crop, scalemix, Asym Aug, Mean Encoder and Sync BN, etc. It would make the paper less precise and overclaim if the terminology of "asymmetric tricks" were not well/ better defined.**
>
> We believe what we call "asymmetric tricks" is clear, e.g., in Sec. 2.1, the first sentence of Sec. 3, etc.
>
>
> **Q7. Minor issues about pseudo-code: 1) misleading naming of standardization layer by "bn". 2) Over complicated argument, i.e. N\_GPUS, and it's not necessarily number of partitions.**
>
> We have revised our pseudo-code in the updated manuscript.

---

### Official Review · Reviewer_w8kU · 2022-07-19

**Rating:** 5
**Confidence:** 4
**Soundness:** 2 fair
**Presentation:** 2 fair
**Contribution:** 3 good

**Summary:**

This paper proposed a new non-contrastive learning algorithm, with a additional variance preserving term in the objective, to get rid of the stop-gradient operation commonly used in non-contrastive learning. They provides some theoretical intuition for their method and empirical comparisons between different ways of preventing representation collapses.

**Questions:**

None

**Limitations:**

The major limitation of this paper is in its flawed explanation of the proposed method, which loosed its validity because (1) the unanswered reason why the theorem in section 3.2 is important for the non-contrastive learning process (with stop-gradient), and (2) why the objective in equation (16) is equivalent to that in (17), or can be motivated by the same theory behind (16).

**Strengths And Weaknesses:**

Strengths:
The empirical comparison is thorough and the ablations are very informative of the effects of each components in the method. They have a (although flawed) theory to back their choice of the new regularizer.

Weaknesses:
1. The major weakness is that the theoretical derivations are not well explained, and do not provide any insights into their proposed method. The results in section 3.2, especially the later comments in equation (14) and line 132-133 only says that with iteration $t$ tends to infinity, the matrix $W_pW_p^{\top}$ will approach $1/n\sum_{i}f_if_i^{\top} $, which can still be a collapsed solution (for example when all $f_i$ are the same and $W_p$ is rank-1). The technical derivations in section 3.2 do not answer why this dynamics (the convergence of $W_pW_p^{\top}$) is needed for the network to avoid collapse, and why it can be utilized to help designing a new covariance regularizer.
2. The transition between equation (16) and (17) is unclear and not well-explained. The transition between (16) and (17) is muddled through with unjustified analogy. This weakens the validity of the explanation in section 3.2 on their final loss form (17) and (19).

---

> ### Author Response · Authors · 2022-08-02
> **Author response to Reviewer w8kU**
>
> **Q1. The major weakness is that the theoretical derivations are not well explained, and do not provide any insights into their proposed method. The results in section 3.2, especially the later comments in equation (14) and line 132-133 only says that when $t\rightarrow\infty$, $W_p^\top W_p$ will approach $\frac{1}{n}\sum_i f_i f_i^\top$, which can still be a collapsed solution. The technical derivations in section 3.2 do not answer why this dynamics is needed for the network to avoid collapse, and why it can be utilized to help designing a new covariance regularizer.**
>
> While why and how BYOL/SimSiam can avoid collapse is the main concern, we are not trying to find a direct answer. We instead try to uncover the connection between BYOL/SimSiam and BT/VICReg. If the two are equivalent, then the understanding of the BT/VICReg's mechanism to avoid collapse will be applicable. Therefore, the comment "(the theoretical derivations) do not provide any insights..." is not true. While we agree that Sec.~3.2 alone does not explain the mecnahism, we think "why (the technical derivations in 3.2) can be utilized to help designing a new covariance regularizer" is clear: replacing an implicit constraint with an explicit constraint.
>
>
> **Q2. The transition between equation (16) and (17) is unclear and not well-explained. The transition between (16) and (17) is muddled through with unjustified analogy. This weakens the validity of the explanation in section 3.2 on their final loss form (17) and (19).**
>
> Although we omitted in the submitted manuscript, the original loss (16) actually works; see table below for the comparison of (16) and (17) in the same setting of the experiments of Sec. 5. We introduce (17) since the loss (16) tends to be more sensitive to the choice of the regularization weight (i.e., $c_\mathrm{cov}$ in (17)); there is no more reason.
>
> Regarding the partitioned loss (19), we want to note that the original loss (16)/(17) can learn good representation, which is enough to support our claimed connection between BYOL/SimSiam and BT/VICReg. We report the loss (19) since we found that it further boosts the performance. No more reason.
>
> In other words, we position these two as optional extensions.
>
>
>
> | Loss | Projector      | #Part | INV | COV  | ACC  |
> |------|----------------|-------|-----|------|------|
> | (16) | 8192-8192-8192 | 8     | 25  | 0.96 | 68.3 |
> | (17) | 8192-8192-8192 | 8     | 25  | 8    | 69   |

---

> > ### Comment · Reviewer_w8kU · 2022-08-09
> > **Thanks for the additional experiments.**
> >
> > After reading the additional results in the rebuttal and the new version of the paper, I decided to raise my score in acknowledgement of the contributions of this paper. Exploring novel approaches of non-contrastive learning is important, and any success in this attempt should be taken seriously.
> >
> > However, since the form of objective (19) used in this paper is closer to the line of papers that use explicit regularizers (which do not require stop-grad), the main contributions of "asymmetry without stop-grad" here is still in doubt (especially the asymmetry here is not absolutely necessary according to Table 3) , and the "bridging" claim is still unsubstantiated.
> >
> > Moreover, I still hold the opinion that the main part of the paper can be presented better, for example, by incorporating more ablations comparing objectives (16), (17) and the final form (19). The main selling point of this paper is that the proposed method is motivated to a large extent by the mathematical derivation, but the emphasis shifted to a weakly related method when reporting the experimental results, which feels a bit unjustified. To me, the objective (16) and (17) worth more considerations.
> >
> > I will leave further discussions to other reviewers, the area chairs and senior area chairs.

---

> > > ### Author Response · Authors · 2022-08-09
> > > **Further author response to Reviewer w8kU**
> > >
> > > “Asymmetry without StopGrad” is not the main contribution. (We assume this term refers to the method having the regularization term, a predictor, and no StopGrad.) Correctly, the main contribution is that we have shown the connection between BYOL/SimSiam (i.e., methods with StopGrad at their core) and decorrelation-based methods that do not require StopGrad. Specifically, we have shown that the former methods transform themselves into the latter by replacing StopGrad with an equivalent regularization. (Note that the regularization term is mathematically derived from the optimization dynamics with StopGrad.) Thus, “asymmetry without StopGrad” is an intermediate form in this transformation. The final form has the regularization term, no predictor (i.e., I-PRED in Table 3), and no StopGrad, which is almost identical to VICReg, as the reviewer states. In short, it shows our success, not failure, that (19) is similar to VICReg. We will further revise the manuscript to make the above clearer.

---

### Author Response · Authors · 2022-08-02
**Author response to all reviewers**

We think that some lack of clarity in our explanation has led to confusion and misunderstanding. To make the study more clear-cut, we first summarize what we have shown in the paper as follows:

- We have theoretically shown (under some assumptions) that minimizing the loss (4) with StopGrad and a predictor (i.e., BYOL/SimSiam) leads to that (a) $W_p^\top W_p \rightarrow F^1F^{1\top}$ as $t\rightarrow \infty$.
- We have shown an empirical observation with BYOL (and many others including ours) that (b) $W_p^\top W_p\rightarrow\mathrm{diagonal}$ with increasing $t$.
- We have proposed a novel method incorporating an explicit regularizer to enforce (a) while removing StopGrad.
- We have experimentally shown that the method works; it learns as good representation as BYOL/SimSiam even though it does not use StopGrad. We have also presented some extensions (Sec. 4.2 and 4.4) that empirically help convergence and better learning.
- If (b) holds, we can regard our method as a decorrelation method like VICReg (and other similar methods).

Remark 1. $W_p^\top W_p\rightarrow\mathrm{diagonal}$ is a necessary condition for the equivalence of the two. Although we have not proven it theoretically, we empirically verify it; we have not seen a case where $W_p^TW_p$ is not diagonal when representation learning goes well. Therefore, we may conclude their equivalence is strongly implied.

Remark 2. Our method needs standardization of features (i.e., $F^1$), and its mechanism is unclear. However, the necessity of feature standardization is neutral to the above discussion. It is independent of the difference of the two, BYOL/SimSiam and BT/VICReg, since SimSiam and BT need BN, whereas BYOL and VICReg do not. It is true, though, that we have another question.

---

### Meta-Review · Area_Chair_D9nL · 2022-08-23

**Recommendation:** Accept
**Confidence:** Certain

**Metareview:**

The paper connects non-contrastive SSL methods (such as BYOL/SimSiam) with feature de-correlation approaches (e.g., Barlow-Twins and VICReg) via theoretical analysis and empirical validations. The paper is well-motivated and the problem to be addressed is important for the community. The experiments seem to be extensive.

There are thorough discussions between authors and reviewers. Reviewers now agree that the paper can be accepted, while the supports remain limited.

There are certain issues that AC strongly advises the authors to address in their camera ready version. One example is to make sure the authors clearly explain which part is proved and which part is observed empirically (e.g., $W_p^\top W_p \rightarrow I$). For empirical observation, please clearly state the experimental setup. AC suspects that this empirical observation ($W_p^\top W_p \rightarrow I$) heavily depends on whether the underlying feature $F$ has been standardized and the high-dimensional output of the projector (i.e., vectors tend to be orthogonal in high-dimensional space).

**Award:**

No

---

### Decision · Program_Chairs · 2022-09-14

Accept